# Vibrational wavepacket dynamics in Fe carbene photosensitizer determined with femtosecond X-ray emission and scattering

Kristjan Kunnus [1*], Morgane Vacher [2], Tobias C.B. Harlang[3,4], Kasper S. Kjær[1,3,4], Kristoffer Haldrup [4], Elisa Biasin[1,4], Tim B. van Driel[5], Mátyás Pápai [6], Pavel Chabera [3], Yizhu Liu[3,7], Hideyuki Tatsuno[3], Cornelia Timm[3], Erik Källman[2], Mickaël Delcey[2], Robert W. Hartsock[1], Marco E. Reinhard [1], Sergey Koroidov[1], Mads G. Laursen[4], Frederik B. Hansen[4], Peter Vester [4], Morten Christensen[4], Lise Sandberg[4,8], Zoltán Németh[9], Dorottya Sárosiné Szemes[9], Éva Bajnóczi [9], Roberto Alonso-Mori [5], James M. Glownia[5], Silke Nelson[5], Marcin Sikorski[5], Dimosthenis Sokaras[10], Henrik T. Lemke [5], Sophie E. Canton [11,12], Klaus B. Møller [6], Martin M. Nielsen [4], György Vankó [9], Kenneth Wärnmark[7], Villy Sundström [3], Petter Persson [13], Marcus Lundberg [2], Jens Uhlig [3] & Kelly J. Gaffney [1*]

The non-equilibrium dynamics of electrons and nuclei govern the function of photoactive materials. Disentangling these dynamics remains a critical goal for understanding photoactive materials. Here we investigate the photoinduced dynamics of the $[Fe(bmip)_2]^{2+}$ photosensitizer, where bmip = 2,6-bis(3-methyl-imidazole-1-ylidine)-pyridine, with simultaneous femtosecond-resolution Fe Kα and Kβ X-ray emission spectroscopy (XES) and X-ray solution scattering (XSS). This measurement shows temporal oscillations in the XES and XSS difference signals with the same 278 fs period oscillation. These oscillations originate from an Fe-ligand stretching vibrational wavepacket on a triplet metal-centered ($^3$MC) excited state surface. This $^3$MC state is populated with a 110 fs time constant by 40% of the excited molecules while the rest relax to a $^3$MLCT excited state. The sensitivity of the Kα XES to molecular structure results from a 0.7% average Fe-ligand bond length shift between the 1 s and 2p core-ionized states surfaces.

[1] PULSE Institute, SLAC National Accelerator Laboratory, Stanford University, Menlo Park, CA 94025, USA. [2] Department of Chemistry - Ångström laboratory, Uppsala University, Box 53875121 Uppsala, Sweden. [3] Department of Chemical Physics, Lund University, P.O. Box 12 422100 Lund, Sweden. [4] Department of Physics, Technical University of Denmark, DK-2800 Lyngby, Denmark. [5] LCLS, SLAC National Accelerator Laboratory, Menlo Park, CA 94025, USA. [6] Department of Chemistry, Technical University of Denmark, Kemitorvet 207, DK-2800 Kongens Lyngby, Denmark. [7] Centre for Analysis and Synthesis, Department of Chemistry, Lund University, P.O. Box 12422100 Lund, Sweden. [8] University of Copenhagen, Niels Bohr Institute, Blegdamsvej 17, 2100 Copenhagen, Denmark. [9] Wigner Research Centre for Physics, Hungarian Academy of Sciences, P.O. Box 49H-1525 Budapest, Hungary. [10] SSRL, SLAC National Accelerator Laboratory, Menlo Park, CA 94025, USA. [11] ELI-ALPS, ELI-HU Non-Profit Ltd., Dugonics ter 13, Szeged 6720, Hungary. [12] Deutsches Elektronen-Synchrotron (DESY), Notkestrasse 85, D-22607 Hamburg, Germany. [13] Theoretical Chemistry Division, Lund University, P.O. Box 12422100 Lund, Sweden. *email: kkunnus@stanford.edu; kgaffney@slac.stanford.edu

Mechanistic understanding of ultrafast nonadiabatic photo-physical and photo-chemical processes requires correlating the coupled dynamics of the many electronic and nuclear degrees of freedom[1,2]. Deeper knowledge of molecular-level dynamics has the potential to inform the development of efficient and cost-effective functional materials. In particular, progress in the synthesis of novel Fe-based molecular photosensitizers[3,4] highlights a continuing need for a better understanding of intramolecular photoinduced electronic excited state processes in these systems[5–7]. A central challenge to developing Fe-based molecular photosensitizers and photocatalysts is controlling the interplay between charge transfer (CT) and metal centered (MC) excited states. Characterizing this interplay has proven difficult for ultrafast optical methods because of the lack of clear spectroscopic signatures of MC excited states, and highlights the strengths of ultrafast X-ray spectroscopy and scattering methods[8,9].

A powerful method to disentangle the electronic and nuclear motions of 3d transition metal complexes during ultrafast nonadiabatic processes is time-resolved 1s-2p (Kα) and 1s-3p (Kβ) X-ray emission spectroscopy (XES)[10–14] combined with X-ray solution scattering (XSS)[15–23]. These experiments have been utilized to assign the electronic and structural motions during excited state dynamics[24–26] and to project the locations of conical intersections between excited state potential energy surfaces (PES) onto critical structural coordinates[27]. Robust tracking of electronic state populations in these studies has relied on the demonstrated sensitivity of the XES spectra to the charge and spin density on the 3d transition metal[28–31] and the presumed insensitivity of Kα/Kβ XES

to bonding geometry. This approach for interpreting time resolved Kβ XES measurements has been effective for initial time-resolved investigations, but the role of nuclear structure on XES spectra remains unclear.

Coherent nuclear dynamics are ubiquitous in photoexcited systems. Oscillatory wavepacket motions have been clearly measured in time-resolved XSS[19,27,32], however the effects of purely nuclear dynamics in ultrafast Kα/Kβ XES has not been observed[10–14]. This contrasts with other X-ray spectroscopic methods that directly involve valence orbitals, such as X-ray absorption[33–38] and resonant inelastic X-ray scattering[39–41], which are sensitive to both electronic and nuclear structure. The development of ultrafast XES requires understanding the impact of nuclear dynamics on Kα and Kβ XES spectra.

This study focuses on the application of ultrafast XES and XSS to photoinduced electronic excited state dynamics in the Fe carbene complex [Fe(bmip)$_2$]$^{2+}$ [bmip = 2,6-bis(3-methyl-imidazole-1-ylidine)-pyridine] (Fig. 1a)[42–44]. The strong σ-bonding of metal-carbene complexes leads to destabilization of MC excited states and this generates uncharacteristically long $^3$MLCT lifetimes for iron complexes[42] and a high electron injection efficiency[45]. These experimental findings have motived theoretical studies focused on understanding the $^3$MLCT excited state relaxation mechanism[46–48]. Unlike the experimental studies, quantum dynamical simulations indicate fast population of the $^3$MC excited states in [Fe(bmip)$_2$]$^{2+}$, with only 1/3 of the excited state population remaining in the MLCT manifold and 2/3 forming a $^3$MC excited state with a ~1 ps time constant[47]. Addressing the discrepancies between the interpretation of

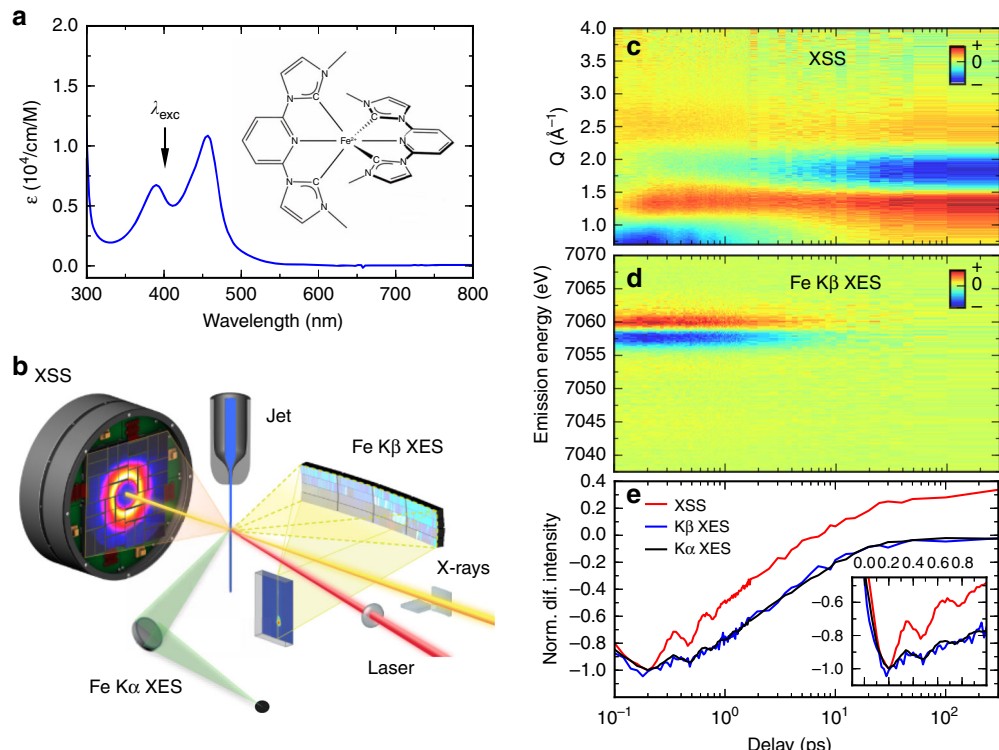

**Fig. 1 Simultaneous tracking of femtosecond dynamics with X-ray scattering and emission. a** UV–visible absorption spectrum of [Fe(bmip)$_2$]$^{2+}$ in acetonitrile (ε—molar absorptivity) and its chemical structure (inset). $\lambda_{exc}$ indicates the excitation wavelength at 400 nm. **b** Schematic of the experimental setup, adapted from ref. [27] with permission from the Royal Society of Chemistry. A jet of the sample solution (blue) is excited with 400 nm laser light (red) and followed by 8.5 keV X-ray probe beam (yellow). XSS is collected with a large area detector downstream, whereas Kα/Kβ XES are detected with two spectrometers positioned close to a 90° scattering geometry. **c** Time-resolved difference signals of XSS (Q—length of the scattering vector), **d** Time-resolved difference Fe Kβ XES spectra. **e** Time-resolved Fe Kα/Kβ XES and XSS traces (black: Kα at 6404.3 eV; blue: Kβ between 7056 and 7058.5 eV; red: XSS between 0.7 and 1.0 Å$^{-1}$). Inset highlights the oscillatory component of the XES and XSS signals. Difference traces normalized to −1 at the minimum (Kβ scaled to maximize overlap with Kα). Source data are provided as a Source Data file.

ultrafast optical spectroscopy and quantum dynamics studies requires robust signatures for MLCT and MC excited states and motivates our time-resolved XES and XSS study of the coupled electronic and structural dynamics of $[Fe(bmip)_2]^{2+}$.

In the present study we demonstrate that the time-resolved Kα XES signal of $[Fe(bmip)_2]^{2+}$ shows the signature of vibrational wavepacket dynamics. We verify the assignment of the oscillations in the Kα XES signal with the simultaneous observation of the same wavepacket dynamics with XSS and assign the oscillations to a Fe-ligand bond stretching coordinate of a $^3$MC excited state. Combining these experimental observations with quantum chemical simulations, we quantify the core-level vibronic coupling responsible for the XES structural sensitivity. These results highlight the importance of vibronic effects in time-resolved XES experiments and demonstrate the role of $^3$MC excited states in the electronic excited state relaxation dynamics of an Fe carbene photosensitizer.

## Results

**Simultaneous femtosecond XES and XSS.** We probed the dynamics of $[Fe(bmip)_2]^{2+}$ dissolved in acetonitrile following 400 nm photoexcitation of a MLCT excited state with ultrafast XSS and Fe Kα/Kβ XES. A schematic of the experimental configuration is shown in Fig. 1b. Three signals were detected simultaneously as a function of pump-probe delay: (1) XSS with a scattering range of 0.5–5.0 Å$^{-1}$, (2) XES over the full Fe Kβ spectral energy range (7035–7070 eV) and (3) intensity at a fixed emitted photon energy of 6404.3 ± 0.2 eV corresponding to the maximum of the Kα$_1$ XES peak (Fig. 1c–d). Unexpectedly, for time delays of <1 ps the Kα XES signal displays a clear oscillatory component, an oscillation potentially present in the Kβ XES as well (Fig. 1e, inset). These oscillations are visible also in the XSS signal and they have been observed previously in UV–visible pump-probe measurements (Supplementary Information of ref. [42], Supplementary Note 7, and Supplementary

Figs. 14 and 15). Both XSS and XES oscillations are in-phase and have the same period, providing clear evidence the oscillations originate from the same dynamical source. In order to comprehensively understand the dynamics underlying these oscillatory features, we proceed by combined quantitative analysis of all three detected signals. Firstly, we will determine the excited-state population dynamics of $[Fe(bmip)_2]^{2+}$ molecules based on Fe Kα/Kβ XES. Secondly, the accompanying structural dynamics is characterized with XSS. Thirdly, the excited-state population and structural dynamics will be combined with quantum chemical calculations of Fe Kα XES.

**Excited-state population analysis of XES.** As mentioned in the introduction, both Fe Kα/Kβ XES have a well-established sensitivity to Fe oxidation and spin state (Supplementary Note 1 and Supplementary Figs. 1 and 2). By comparing the shape of the measured Kβ XES difference spectrum with the spectra of various Fe complexes from previous studies[10,14], we can immediately exclude high spin 3d$^6$ or 3d$^5$ excited state configurations because of the lack of a Kβ' feature at 7045 eV (Fig. 2a). This distinguishes $[Fe(bmip)_2]^{2+}$ from an analogous complex with t-butyl side-groups on the ligands where the high spin 3d$^6$ excited state forms following MLCT excitation[49]. Instead, the dominant Kβ XES difference signal shows a blue shift of the main Kβ$_{1,3}$ peak at 7058 eV. Such a shift with respect to the low-spin Fe 3d$^6$ ground state is consistent with $^1$MLCT/$^3$MLCT excited states, as both form a low-spin 3d$^5$ doublet from the Fe perspective[11] or with a 3d$^6$ intermediate spin $^3$MC excited state[14]. This is evident also from the comparison with $S = 0.5$ and $S = 1$ model complexes in Fig. 2a ($S$—nominal spin of the Fe). Note, the difference spectra generated by the subtraction of the Fe singlet ground state from the Fe doublet and triplet configurations have similar spectral shapes, but the $S = 1$ XES difference spectrum amplitude is about twice the amplitude of the $S = 0.5$ difference spectrum. Because

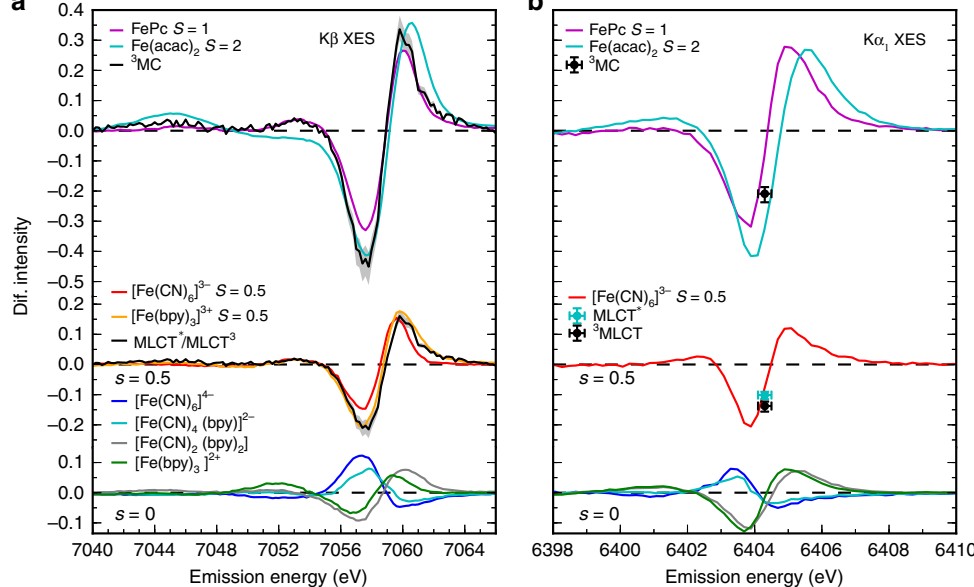

**Fig. 2 Fe Kα/Kβ XES electronic state and ligand dependence. a** Kβ and **b** Kα XES difference spectra ($S$—nominal spin of the Fe center; bpy = 2,2′-bipyridine, Pc = phtalocyanine, acac = acetylacetone). (Lower level) XES difference spectra of model $S = 0$ complexes with various ligands (blue, cyan, gray and green lines). (Middle level) Kβ XES difference spectra (black lines) and Kα difference intensities (black/cyan dots) of MLCT ($S = 0.5$) excited states as retrieved from the XES population kinetics fitting described in the main text, compared with the $S = 0.5$ model spectra (red and orange lines). (Upper level) Similar comparison for the $^3$MC ($S = 1$) $[Fe(bmip)_2]^{2+}$ spectra/intensities and the $S = 1$ (magenta lines) and $S = 2$ (cyan lines) model spectra. Shaded areas (**a**) and error bars (**b**) correspond to the fits with excitation yield within the range of 74–94%. Difference spectra are calculated by subtracting the ground state spectrum of $[Fe(bmip)_2]^{2+}$ ($S = 0$). Complete model spectra are included in the Supplementary Note 1.

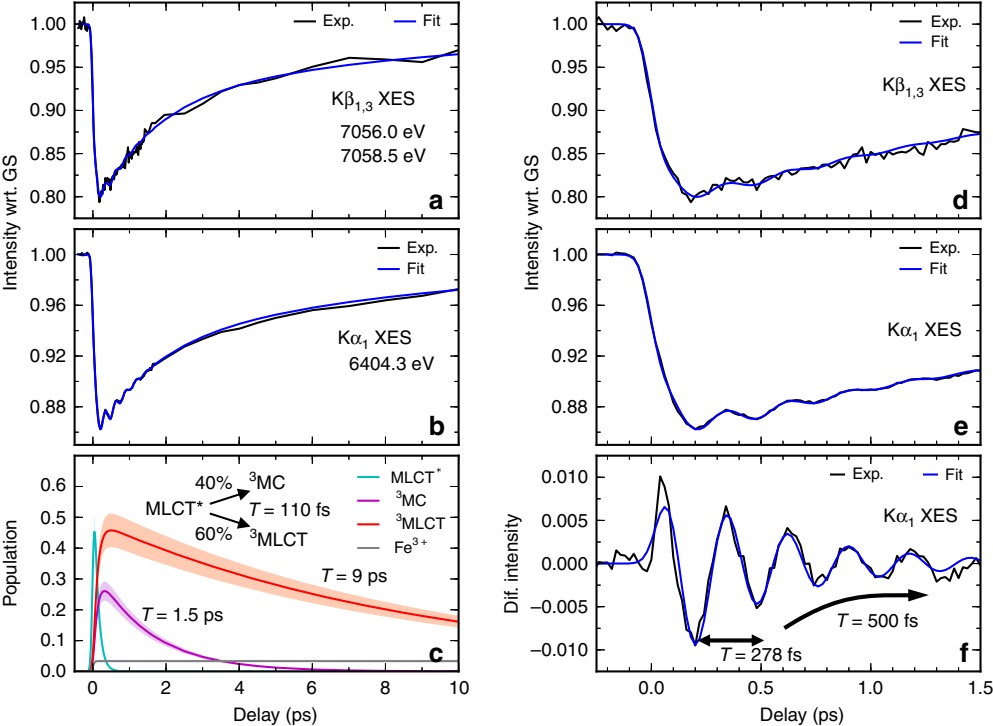

**Fig. 3 Combined analysis of the time-resolved Kβ and Kα XES data. a**, **b** Fits of XES time dependence with the three-state branching model discussed in the main text (data: black lines; fits: blue lines). **c** Electronic excited states populations derived from the fit with 84% excitation yield (MLCT*: cyan; $^3$MC: magenta; $^3$MLCT: red; $Fe^{3+}$: gray). Shaded areas correspond to an excitation yield range of 74–94%. Inset shows the derived population kinetics model: MLCT* decays with 110 fs by branching into the $^3$MLCT (60%) and into the $^3$MC (40%) states. $\tau$ is the lifetime of the respective transient excited states ($^3$MC: 1.5 ps; $^3$MLCT: 9 ps). **d**, **e** Fits of early time dynamics exhibiting oscillatory signal. **f** Oscillatory Kα XES signal after subtraction of the non-oscillatory part. $T$ is the oscillation period and $\tau$ is the lifetime of the oscillations (damping constant).

the XES difference spectra of these states effectively differ only by their amplitude, assignment between these two requires the quantitative analysis of the Kα/Kβ XES presented below.

The Kβ XES difference signal does not show any significant time dependent spectral shape changes (Fig. 1d, Supplementary Note 3 and Supplementary Figs. 5 and 6). Therefore, we focus here on the analysis of the spectral intensities at the region of biggest difference signal between 7056 and 7058.5 eV, together with the Kα XES intensity recorded at 6404.3 eV (Fig. 3). The time-dependence of the Kα XES intensity follows almost exactly the Kβ XES dynamics. Close inspection of both signals reveals that the excited state population dynamics cannot be described by a single excited state (Supplementary Note 9 and Supplementary Figs. 21–24). Firstly, both XES time traces show an exponential rise in amplitude, demonstrating that at least a fraction of the initial photoexcited MLCT state transitions to a state with a larger absolute difference signal on the 100 fs timescale. Secondly, the relaxation dynamics to the electronic ground state are biexponential, with the faster decrease in the intensity decaying with a 1–2 ps time constant, followed by a slower decay with a ~10 ps time constant. We have constructed a kinetic model capable of capturing all these features. Within this model, the initially photoexcited MLCT* state branches into a $^3$MC and a long-lived MLCT state, which subsequently decay with different time-constants to the electronic ground state (similar kinetics were recently also observed in another Fe-carbene complex[50]). Based on the absence of stimulated emission, prior UV–visible pump-probe measurements have assigned the long-lived MLCT to a $^3$MLCT (note, $^1$MLCT and $^3$MLCT have the same Fe XES spectra)[42]. Using this kinetic model, we carried out simultaneous fitting of the Kα and Kβ XES traces (Supplementary Note 16). We did not impose any constraints on

the MLCT*, $^3$MLCT, $^3$MC, or the ground state (GS) relative intensities, except for requiring the MLCT* and $^3$MLCT Kβ intensities to be equal. The excitation yield was fixed to 84 ± 10%, determined from a reference sample in identical laser fluence conditions (see Supplementary Note 8 and Supplementary Figs. 16–20). At this high excitation yield, analysis of the XES data at time delays >100 ps revealed 4% of the excited molecules do not decay back to the GS ($Fe^{3+}$ population in Fig. 3c resulting from two-photon ionization of $[Fe(bmip)_2]^{2+}$). However, all other aspects of the kinetic model used to fit the XES signal can be used to fit optical transient absorption data over a range of lower fluence, demonstrating the high optical laser fluence does not transform the relaxation dynamics of $[Fe(bmip)_2]^{2+}$, as discussed in detail in the Supplementary Notes 5 and 6 and Supplementary Figs. 8–13. The XES fitting found that the MLCT* decays with 110 ± 10 fs time constant with 40 ± 10% $^3$MC yield and 60 ± 10% $^3$MLCT yield. Subsequently both states decay into the GS, with time constants of 1.5 ± 0.5 ps and 9 ± 1 ps, respectively (Fig. 3c). A comparison of the extracted XES intensities of the transient states with the various Fe model complexes confirms the assignment of the $^3$MLCT and the $^3$MC states (Fig. 2). The change in the XES difference signal for the $^3$MC state is approximately twice the difference signal associated with the MLCT states, although the intensities of the extracted transient $^3$MC and MLCT XES spectra do not match exactly with the model spectra. This can be readily explained by the 5–10% changes in the XES intensity due to different ligand environments, as seen for the ground state spectra of different low-spin Fe polypyridyl and cyanide complexes with $S = 0$ and $S = 0.5$ spin states (Fig. 2). Additionally, the $^3$MC and the $^3$MLCT states have distinct XSS signals due to their different structure. In the next section, we will show that the XSS dynamics also require

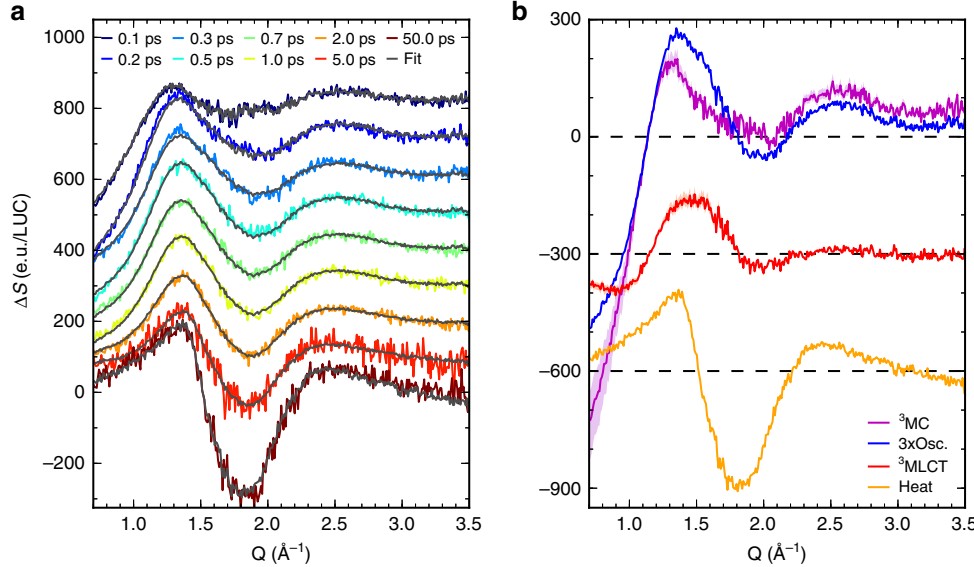

**Fig. 4 Global fit of the time-resolved isotropic XSS data based on the SVD. a** Difference XSS scattering signals (colored lines) and fits (black lines) at selected time delays (offset step 100 e.u./LUC). **b** Difference XSS signals of the time-dependent components derived from global analysis: $^3$MC scattering signal (magenta line), oscillatory signal (blue line, scaled by factor of 3), $^3$MLCT signal (red line) and MeCN heating signal (orange line). All signals are normalized to electronic units per liquid unit cell (e.u./LUC).

population of two states with different Fe-ligand bond lengths and lifetimes, consistent with the $^3$MC and the $^3$MLCT states and the kinetic model derived from the XES.

Simultaneous with the excited state population analysis, we fitted the oscillatory signal. This signal is described by an exponentially damped harmonic oscillation, convolved with the instrument response function (100 fs FWHM) and the 110 fs exponential decay of the MLCT*. Fitting yields a period of $278 \pm 2$ fs, with a damping constant of $500 \pm 100$ fs (Fig. 3f). The K$\beta$ XES spectra show very weak and inconclusive oscillatory dynamics (Supplementary Note 4 and Supplementary Fig. 7). Given the order of magnitude worse signal to noise for K$\beta$ XES, compared to the K$\alpha$ signal, the absence of a clear oscillatory signal may predominantly result from a difference in the count rates for K$\alpha$ and K$\beta$ emission due to the larger K$\alpha$ cross-section and the differences in the K$\alpha$ and K$\beta$ spectrometer designs.

**XSS global analysis.** Time-resolved XSS signals from solutions can be categorized into two groups: (1) solute related dynamics, including changes in the solute and the solute-solvent pair-distribution functions and (2) bulk solvent structure, typically described by changes in temperature and density[51,52]. The XSS difference signal for time delays following solute electronic excited states relaxation are dominated by the latter contribution. Specifically, the observed broadening of the bulk acetonitrile (MeCN) scattering feature at 1.8 Å$^{-1}$ after time delays > 5 ps corresponds to the heating of the solvent (Fig. 1c)[53]. However, for the purposes of the present study, we focus on the structural dynamics associated with changes in the solute and solute-solvent pair distribution functions. Within 100 fs, a strongly negative difference scattering signal appears below 1.0 Å$^{-1}$, as well as a weakly positive signal between 1.25 and 1.75 Å$^{-1}$ (Fig. 1c). The negative low-Q signal indicates expansion of the Fe-ligand bond lengths, similar to previously observed XSS signals associated with the elongated metal-ligand bond lengths of MC excited states in polypyridine metal complexes[19,23,25]. Additionally, this XSS signal exhibits the same dynamical behavior observed in the Fe K$\alpha$/K$\beta$ XES difference signals. In particular, the solute XSS signal decays to the ground state biexponentially, consistent with the population of two excited

states. However, because of the overlapping of solute and solvent XSS contributions, direct comparison of the XES and XSS difference signals is difficult. To facilitate this comparison, and to separate the XSS signal into components with distinct dynamics, we carry out a global analysis of the time-resolved XSS data.

Global analysis of the XSS difference data in Fig. 4 is based on the electronic state populations derived from the XES and the singular value decomposition (SVD) of the two-dimensional XSS data[54]. We fix the $^3$MC and MLCT populations and the oscillatory component to be exactly the same as found from the fitting of K$\alpha$/K$\beta$ XES (Fig. 2). This facilitates assignment of the structural signals to specific electronic states and ensures consistency with the XES analysis, without imposing any functional form of the related Q-dependent XSS signals. Since the difference scattering components used in the global analysis come from the SVD of the XSS data, this does not represent a fit to a specific structural model for the solute or the solute-solvent pair distribution function. The time-dependent component needed to describe heating of the MeCN solvent is retrieved through the following procedure. The Q-dependent XSS solvent heating signal is taken to be equal to the average XSS difference signal between 50 and 500 ps, a time range where the MLCT and $^3$MC solute related signals have vanished (the photoionized solute signal is by a factor of 100 smaller than the heating signal and therefore the late XSS signal accurately describes hot solvent). Subsequently, by fitting the amplitude of this signal at each time delay, we find that the increase of solvent temperature can be described by two parallel exponential time constants of 0.35 ps and 13.2 ps (Supplementary Note 11 and Supplementary Figs. 26–31). With these four time-dependent components it is possible to accurately fit the dynamics observed for >200 fs time delays in the XSS. We find that the <200 fs structural dynamics cannot be captured by the population dynamics of the MLCT* due to both solvent and solvation contributions to the difference signal that we have not included in the structural model (Supplementray Note 11). However, these early dynamics do not influence the analysis of the solute dynamics at later times.

The extracted XSS signals associated with the individual dynamical components are displayed in Fig. 4b. Firstly, the XSS

**Table 1 Calculated distances between Fe and the coordinating atoms in the relevant [Fe(bmip)₂]²⁺ electronic states from ref. [46].**

| Electronic state | Fe-C (Å) | Fe-N (Å) | Fe-L[a], R (Å) |
|---|---|---|---|
| GS | 1.952 | 1.924 | 1.943 |
| $^3$MLCT | 1.984 | 1.919 | 1.962 |
| $^3$MC (short bond length) | 1.977 | 2.062 | 2.066 |
| $^3$MC (long bond length) | 2.087 | 2.205 | |

[a]Fe-L refers to the average bond length between the Fe and the coordinating atoms (C and N) of the ligands.

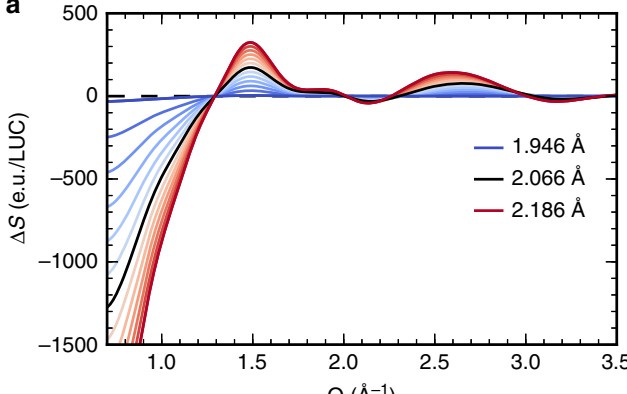

signal of the $^3$MC state below 1 Å⁻¹ is about 6–8 times more negative than the $^3$MLCT signal. This is consistent with significantly elongated metal-ligand bond lengths of the $^3$MC state (~0.1 Å increase), whereas the $^3$MLCT excited states metal-ligand elongation is significantly smaller (~0.01 Å increase). We verify this by simulating the $^3$MC XSS signal in the next section and confirm the assignment identified in the XES analysis. Secondly, the shapes of the XSS signals associated with the $^3$MC population and the oscillations are rather similar (Fig. 4b). This is a strong indication that the oscillations originate from structural dynamics on the $^3$MC PES. In addition, given the small amplitude of the $^3$MLCT XSS signal, it is not possible that the observed oscillations originate from nuclear wavepacket dynamics on the $^3$MLCT PES. We, therefore, conclude that the observed oscillations originate from the dynamics on the $^3$MC PES and we proceed with a quantitative modeling of these structural motions.

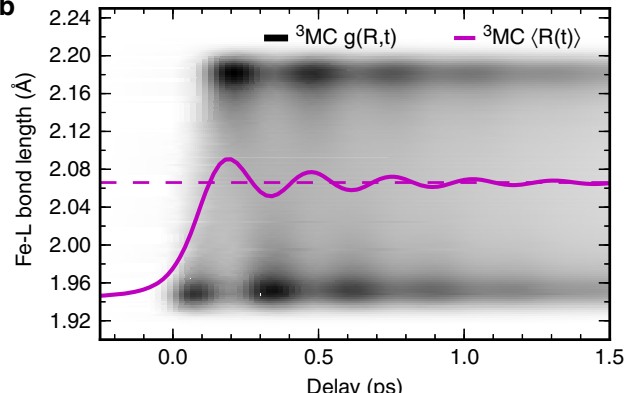

**Simulation of the $^3$MC structural dynamics.** In order to simulate the experimental $^3$MC XSS signals we utilize the published molecular structures[46]. As expected, both Fe-C and Fe-N bond lengths are significantly elongated for the $^3$MC state and the average Fe-ligand bond length is increased by $\Delta R = 0.123$ Å (Table 1). In addition, there is a noticeable pseudo-Jahn-Teller effect in the $^3$MC, resulting in different Fe-ligand bond lengths for the bmip ligands. The dependence of the simulated difference XSS signal from the average Fe-ligand distance in an $R$ range from 1.94 to 2.19 Å is shown in Fig. 5a. The simulation includes changes in the structure of the solute and in the solute-solvent pair-distribution functions (the so-called solvent cage term). The contribution of the solute was calculated by modifying the structure of [Fe(bmip)₂]²⁺ along the coordinate connecting the optimal GS and $^3$MC structures (the GS-$^3$MC coordinate). XSS signals from these structures were calculated using the Debye scattering equation. The effect of the solvent cage was quantified with a molecular dynamics simulation of the optimal GS and $^3$MC [Fe(bmip)₂]²⁺ structures embedded in MeCN (Supplementary Note 13 and Supplementary Fig. 33). Solute-solvent radial distribution functions retrieved from this molecular dynamics simulation were then used to calculate the XSS solvent cage signal[55]. Following Biasin et al.[19], we assume a solvent cage scattering signal proportional to $\Delta R$. This establishes a framework for fitting the difference scattering signal associated with the $^3$MC excited state over the full range of Fe-ligand distances relevant for the dynamics.

In order to simulate the XSS signal of a nuclear wavepacket moving on the $^3$MC PES, we calculated the distribution $g(R,t)$ of a Gaussian wavepacket on a harmonic PES with a vibrational period of 278 fs and a minimum at the $^3$MC geometry $R = 2.066$ Å (Fig. 5b, Supplementary Note 17). The $^3$MC state is populated with 110 fs time constant at the location of the ground state geometry ($R = 1.943$ Å) with zero velocity. We assign the damping of the coherent wavepacket motion, characterized by a 500 fs

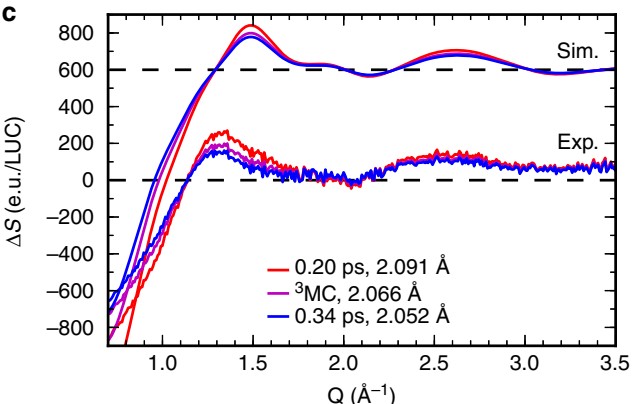

**Fig. 5 Simulated solute difference XSS signals. a** Dependence of [Fe(bmip)₂]²⁺ $^3$MC difference XSS signal from the average Fe-ligand (L) bond length (including solute-solvent scattering). Black line: optimal $^3$MC structure. Blue lines: shorter bond lengths; red lines: longer bond lengths. Respective bond lengths of the curves are from $R = 1.946$ Å to $R = 2.186$ Å, step size is 0.02 Å. **b** Simulated $^3$MC bond lengths distribution $g(R,t)$ as a function of time (white-black color scale) and the corresponding ensemble average Fe-ligand bond length (magenta solid line). Magenta dashed line: optimal (equilibrium) $^3$MC bond length. **c** Simulation of the time-dependent $^3$MC difference signal and comparison to the experimental $^3$MC signal extracted from the global fit (including the oscillatory component). Red line: maximum $^3$MC structural expansion at 0.20 ps (<R> = 2.091 Å) Magenta line: optimal $^3$MC geometry (<R> = 2.066 Å). Blue line: maximum $^3$MC structural contraction at 0.34 ps (<R> = 2.052 Å).

time constant, primarily to dephasing caused by quasi-elastic scattering. Likely, the intramolecular vibrational energy distribution also contributes to the damping, but vibrational cooling to the solvent should not be a dominant cause of damping because

vibrational cooling occurs on the few-to-many picosecond time scale in MeCN. Note however, that the observed XSS signal is primarily sensitive to the ensemble averaged Fe-ligand distance $\langle R(t) \rangle$ and not to the actual distribution of structures (Supplementary Note 12 and Supplementary Fig. 32). Comparison of the resulting simulated $^3MC$ difference XSS signals with the experimental $^3MC$ signal extracted from the global fit shows qualitative agreement (Fig. 5c) and the observed vibrational period agrees well with a calculated low-frequency normal-mode vibration involving Fe-ligand stretching[47]. At present, we do not have a robust model for changes in the solute-solvent pair distribution function. This precludes the quantitative extraction of the Fe-ligand bond length from the XSS signal, but does not preclude the qualitative conclusion that significant Fe-ligand bond expansion and oscillation does occur, consistent with motion on the $^3MC$ excited state PES[46,47]. We, therefore, conclude that the observed coherent structural dynamics can be described by a harmonic Gaussian wavepacket motion along an Fe-ligand stretching coordinate on the $^3MC$ PES.

**Origin of the XES sensitivity to nuclear wavepacket dynamics.** We carried out ab initio Restricted Active Space Self-Consistent Field (RASSCF) calculations of the $^3MC$ excited state Kα XES spectra as a function of average Fe-ligand bond length $R$ to establish the XES sensitivity to the structural dynamics of the molecule. Kα XES spectra of four $[Fe(bmip)_2]^{2+}$ molecular structures were calculated, corresponding to the optimal ground state ($R = 1.943$ Å) and the $^3MC$ ($R = 2.066$ Å) geometries, as well as for one elongated ($R = 2.092$ Å) and one contracted ($R = 2.050$ Å) $^3MC$ structure (Fig. 6a, only two spectra are shown for clarity). We find that the structural effect can be described by an energy shift of the spectrum, without any significant change to the shape. The shift of the spectrum is linear with respect to the Fe-ligand distance and has a magnitude of $\Delta E/\Delta R = 0.3$ eV/ 0.123 Å (corresponding to a change from GS to $^3MC$ geometry for the $^3MC$ electronic state). At the emitted photon energy where the Kα XES data was recorded (6404.3 eV), these calculations indicate that such Fe-ligand bond elongation leads to a linear decrease in the $^3MC$ intensity with a slope of $\Delta I/\Delta R = 10\%/0.123$ Å (relative to the ground state intensity, Fig. 6b). By considering the linear intensity dependence at 6404.3 eV, we can simulate the Kα XES signal based on the $^3MC$ nuclear wavepacket dynamics ($g(R,t)$ in Fig. 5b). We keep all the other populations and relative intensities fixed at the values found from the XES fitting (Fig. 3), except a small decrease in the MLCT* intensity to compensate for the increase in the $^3MC$ intensity due to the XES red shift for shorter Fe-ligand distances. We find that agreement with the experimental Kα XES time trace is achieved if $\Delta I/\Delta R = 16\%/0.123$ Å (Fig. 6c). This structural sensitivity is in reasonable agreement with the value derived from the RASSCF simulations. Additionally, we note that the ligand environment dependent Kα and Kβ spectra of cyanide/bipyridine $S = 0/S = 0.5$ model complexes exhibit the same trend (Fig. 2). The dominant ligand dependent effect is an overall shift of the spectrum, and weakening of the ligand field shifts the spectrum to higher emission energies. Similarly, the magnitude of the effect close to the maximum of the spectrum is ~10%.

## Discussion

We have attained a detailed picture of coupled electronic and nuclear structural dynamics during photoinduced electron transfer in an Fe carbene, $[Fe(bmip)_2]^{2+}$, photosensitizer using the complementary sensitivities of XES and XSS. We found that the directly photoexcited MLCT* state follows two distinct

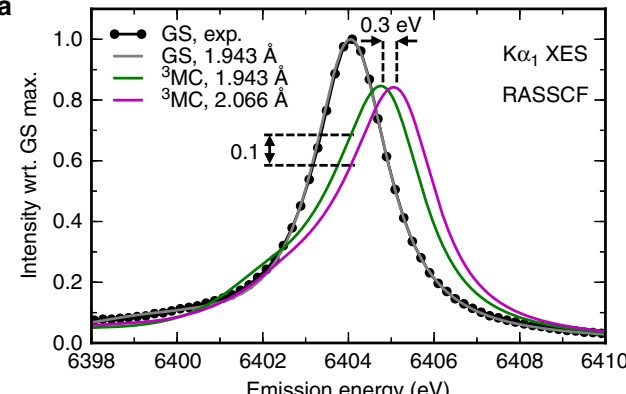

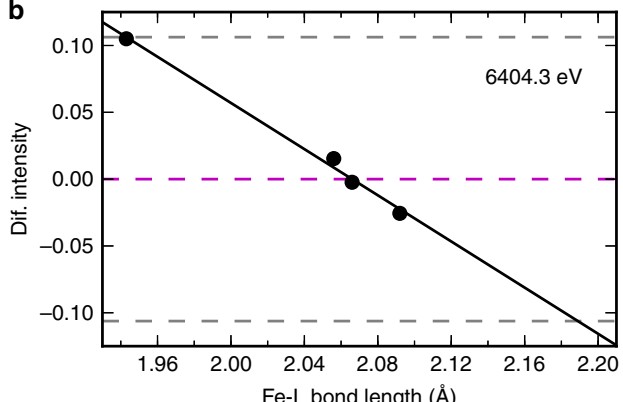

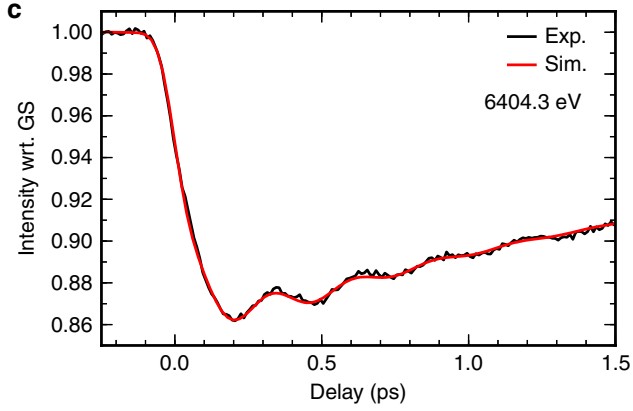

**Fig. 6 Simulated Kα XES signals. a** Experimental XES spectra of the $[Fe(bmip)_2]^{2+}$ ground state (GS, black dots), compared with the calculated GS spectrum (gray line) and the $^3MC$ excited state spectra at the GS geometry (green line) and at the $^3MC$ geometry (magenta line). **b** Calculated $^3MC$ state XES intensity change at 6404.3 eV as a function of the Fe-ligand (L) bond length, relative to the optimal $^3MC$ structure (black dots; black line: linear fit). Dashed lines correspond to the intensity changes at optimal (magenta) and extremal (gray) bond lengths, $R = 2.066 \pm 0.123$ Å. **c** Simulation of Kα XES time-dependent intensity at 6404.3 eV (black line: experiment; red line: simulation).

relaxation pathways (Fig. 7, lower half). 60% of the population relaxes to a $^3MLCT$ excited state and decays with a 9 ps time constant. About 40% of the MLCT* population relaxes to a $^3MC$ state via ultrafast non-adiabatic back-electron transfer to Fe. This $^3MC$ state decays to the ground state with a 1.5 ps time constant. The observation of a 9 ps $^3MLCT$ state lifetime confirms the previous assignment based on the UV–visible transient

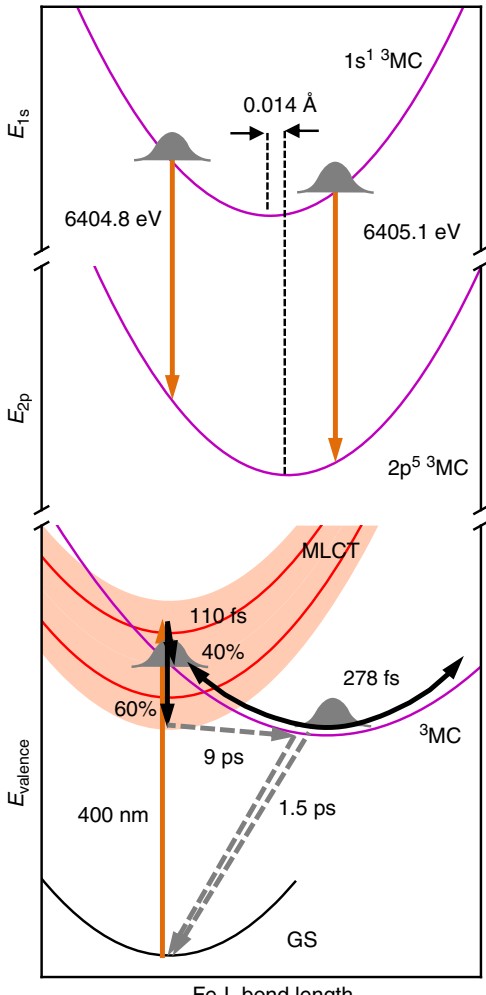

**Fig. 7 Excited state dynamics of [Fe(bmip)$_2$]$^{2+}$.** (Lower half) Population and structural dynamics following photoexcitation. Initial MLCT state relaxes ultrafast with 110 fs time constant and 40% yield to the $^3$MC state which triggers vibrational wavepacket dynamics along an effective Fe-ligand (L) stretching coordinate with 278 fs period. The remaining 60% of the MLCT decays to the GS through the $^3$MC with a 9 ps time constant. Black line: GS potential energy surface (PES); red lines and red shaded area: MLCT states PESs; magenta line: $^3$MC state PES. Orange arrows: radiative transitions; black arrows: fast coherent dynamics; Gray dashed arrows: slower incoherent dynamics. (Upper half) The origin of XES structural sensitivity due to core-level vibronic coupling between 1s and 2p core-ionized states. Note the displacement of the respective $^3$MC 1s and 2p core-ionized PESs (magenta lines). All PESs are qualitative.

absorption experiments with a 485 nm excitation wavelength, although the UV–visible experiments did not detect any $^3$MC population[42]. Importantly, when the complex is excited at 400 nm and probed in the UV–visible, the data can be fit with the same kinetic parameters used to model the XES data (Supplementary Note 6). This supports the conclusion that the branching ratio between $^3$MLCT and $^3$MC formation depends on the pump photon energy.

Recently, similar excited state relaxation dynamics were observed in a related heteroleptic [Fe(bpy)(btz)$_2$]$^{2+}$ complex [bpy = 2, 2'-bipyridine, btz = 4,4'-bis(1,2,3-tri-azol-5-ylidene)][50]. It was determined for this complex that the $^3$MLCT state decay is mediated by the $^3$MC state. Because [Fe(bmip)$_2$]$^{2+}$ has a slightly longer $^3$MLCT and a slightly shorter $^3$MC lifetime than [Fe(bpy)(btz)$_2$]$^{2+}$, we cannot experimentally distinguish between $^3$MLCT→$^3$MC→GS or

$^3$MLCT→GS relaxation pathways here (Supplementary Note 9). Given the otherwise similar population dynamics in these complexes, it is likely that the $^3$MC state facilitates the $^3$MLCT decay also in [Fe(bmip)$_2$]$^{2+}$. Ultrafast $^{1,3}$MLCT→$^3$MC population transfer has been observed before in Fe$^{2+}$ coordination complexes[10,14], but only in iron carbenes has ultrafast $^3$MC formation co-existed with a long-lived $^3$MLCT state. Intuitively, this would require that the MLCT* is resonant with the $^3$MC state, whereas the $^3$MLCT is lower in energy with a small barrier between the $^3$MLCT and $^3$MC. In Fig. 7 we show qualitative PESs consistent with this explanation. There, 400 nm light excites the molecule high in the MLCT manifold. The subsequent branching is likely governed by a competition between electron transfer to the $^3$MC and relaxation within the MLCT manifold. Coupling to different MC states, as well as the initial excitation wavelength could influence the branching ratio[56]. Recently, a branching between the $^3$MC and $^3$MLCT populations has been predicted for [Fe(bmip)$_2$]$^{2+}$ by quantum dynamical simulations, although the simulated time constant of the $^3$MC formation was ~1 ps[47]. This branching between $^3$MLCT and $^3$MC states will influence photosensitizer performance. Iron carbene photosensitizers with carboxalate-functionalized bmip ligands has demonstrated a charge injection efficiency of 92% to TiO$_2$ nanoparticles[45]. Interestingly, this electron injection yield exceeds the $^3$MLCT yield we observe for [Fe(bmip)$_2$]$^{2+}$ dissolved in acetonitrile. Given the expected sensitivity of the branching ratio to small changes in the potential energy landscape, the higher yield for charge injection could result from the red shifted MLCT energies of the COOH-functionalized Fe carbene photosensitizer, differences in the solvation response[57], and the potential for electron injection of electrons from $^3$MC excited states. In summary, the observed population dynamics highlight the critical role of $^3$MC excited states in the MLCT relaxation of [Fe(bmip)$_2$]$^{2+}$.

The time-resolved XSS analysis for the $^3$MC excited state shows Fe-ligand bond expansion consistent with an average Fe-ligand bond length increase of $\Delta R = 0.123$ Å predicted by theory[46]. Since the electron transfer from MLCT* to the $^3$MC excited state occurs with a 110 fs time constant, impulsive compared to the 278 fs period of Fe-ligand stretching mode, relaxation from the MLCT* excited state to the $^3$MC state generates a vibrational wavepacket along this coordinate. Periodic modulation of the Fe-ligand bond length on the $^3$MC PES are clearly captured with XSS. Additionally, these nuclear dynamics lead to oscillations in the Kα XES signal. Based on an ab initio RASSCF calculation we can establish that this structural sensitivity results from a linear shift of the XES spectrum as a function of the Fe-ligand bond length. The linear change in energy is a manifestation of first-order vibronic coupling between the core-levels, resulting in a relative displacement of the 1s and 2p core-ionized PESs along the coordinate of the $^3$MC wavepacket motion (Fig. 7, upper half). The calculated RASSCF core-ionized PESs show noticeable shortening and stiffening of the Fe-ligand bond in comparison to the $^3$MC state without a core-hole. This is due to stabilization of the Fe 3d levels upon the creation of the core hole, which leads to improved overlap with the occupied ligand orbitals. Most importantly, this core-hole induced electronic relaxation is slightly different for 1s and 2p core-holes. Even though the force constant $k_{core}$ of the Fe-ligand stretching coordinate in the presence of either a 1s or 2p core-ionization is the same, the calculated optimal bond length is longer for the 2p state, with displacement $\Delta R_{core} = 0.014$ Å (Fig. 7 and Supplementary Note 15 and Supplementary Figure 34). It follows that the energy difference between the core-ionized PESs changes with a slope of $\kappa_{core} = k_{core}\Delta R_{core}$ (Supplementary Note 18). Here $\kappa_{core}$ is the first-order vibronic coupling constant of the (2p$^5$)$^3$MC states with respect to the (1s$^1$)$^3$MC states[58]. Based on the calculated Kα XES shift, we find that $\kappa_{core} = 2.44$ eV/Å. It is

relevant to compare this dynamic shifting effect to a text-book static broadening effect of vibronic coupling. Considering that the calculated core-ionized PESs vibrational period is 241 fs (hv = 17 meV), the corresponding Huang-Rhys parameter between the core-levels is $S_{core} \approx 1$ (Supplementary Note 18). Because the 1s core-hole lifetime is <1 fs, we can neglect any nuclear dynamics induced by the core-hole. Therefore, one can estimate the vibronic broadening to be 40 meV (FWHM). In contrast, the overall Kα₁ XES broadening is equal to about 1.55 eV (FWHM), dominated by the natural lifetime of 1s and $2p_{3/2}$ states. It is evident that such a small vibronic broadening relative to the lifetime broadening is not detectable in the width of a static Kα XES spectrum. Importantly, one is significantly more sensitive to this vibronic coupling in a time-resolved experiment, because it appears as a shift of the XES spectrum and because the wavepacket samples a wide range of bond lengths.

Prior investigations of the influence of metal-ligand bonding on Fe Kα/Kβ XES spectra have concluded that the spectra change minimally with changes in ligand field[59,60] and exhibit small bonding dependent chemical shifts because both the initial and final state involved in X-ray emission contain a core-hole[61]. The strong iron-ligand coordination bonds present in [Fe(bmip)₂]²⁺ and cyano containing Fe complexes clearly present exceptions, with Kα and Kβ mainline emission peak positions exhibiting noticeable ligand dependent shifts, as shown in Fig. 2. These shifts reflect the significant variation in Fe-ligand bonding for these complexes. Specifically, [Fe(CN)₆]⁴⁻ has a 10Dq value > 1 eV greater than [Fe(bpy)₃]²⁺ and the covalency of the $t_{2g}$ orbitals is significantly larger (~25% compared to ~15%)[62]. For [Fe(bmip)₂]²⁺, the ³MC excited state in the ground state geometry has a 10Dq ~1 eV more than in the optimal geometry[46,47]. As expected, this is related to significant variation in the covalency as a function of Fe-ligand bond length, with the ligand character of the $e_g$ orbitals decreasing from 20% for the ground state geometry to 10% for the equilibrium ³MC geometry (Supplementary Note 14 and Supplementary Table 1). These changes in Fe-ligand interaction result in differential screening of the 1s and 2p core holes as a function of Fe-ligand bond length and different optimal Fe-ligand bond lengths for the 1s and 2p core-ionized states (core-level vibronic coupling).

The magnitude of the core-level vibronic coupling in [Fe(bmip)₂]²⁺ is likely relatively large due to the strong bonding between Fe and the carbene ligands. However, the amplitude of the ensemble averaged <R(t)> oscillations are also significantly suppressed due to the fact the vibrational wavepacket is generated by the 110 fs lifetime of the MLCT* excited state, not through optical excitation of Franck-Condon active vibrational modes. Direct photoexcitation to a displaced PES would enhance the XES oscillations, possibly making the core-level vibronic effect observable also in complexes with weaker metal-ligand bonds.

## Methods

**Experiment**. Polycrystalline [Fe(bmip)₂](PF₆)₂ was synthesized by mixing the imidazolium salt solution with FeBr₂ in cooled potassium tert-butoxide and tetrahydrofuran[42]. The time-resolved XES and XSS measurements were conducted at the X-ray Pump-Probe (XPP) endstation at the Linac Coherent Light Source (LCLS)[63]. The sample consisted of 20.3 mM [Fe(bmip)₂](PF₆)₂ dissolved in anhydrous acetonitrile. A recirculating 50 μm diameter round Rayleigh jet in He environment was used for sample delivery. The sample was excited with 400 nm optical laser pulses of 45 fs duration, 120 μm focus diameter (FWHM), and 45 mJ/cm² fluence. 8.5 keV unmonochromatized X-ray laser pulses of 30 fs duration and 10 μm focal diameter were used to probe the sample. The Fe Kβ XES was spatially dispersed using a von Hamos spectrometer consisting of four dispersive Ge(620) crystal analyzers with a central Bragg angle of 79.1 degrees[64], and detected with a 140k Cornell-SLAC Pixel Array Detector (CSPAD)[65]. Misalignment of an analyzer crystal was corrected with a procedure described in the Supplementary Note 2 and Supplementary Figs. 3 and 4. The Fe Kα XES was analyzed using a spherically bent Ge(440) crystal in Rowland geometry and detected with a second 140k CSPAD. The XSS was detected with 2.3 M CSPAD in forward scattering geometry. The relative timing between pump and X-ray probe pulses was measured

for each X-ray shot[66]. The full 2D images of the XES and XSS detectors were read out shot-to-shot and subsequently processed and binned after the pump-probe delay. 1D XSS scattering curves were retrieved from the 2D CSPAD images after appropriate masking and angular integration[67], including separation of the iso-tropic and anisotropic scattering components (Supplementary Note 10 and Supplementary Fig. 25)[68]. XES emission energies were calibrated to match ground state spectra measured at the Stanford Synchrotron Radiation Lightsource (SSRL). XSS Q-range was calibrated with the acetonitrile heating signal[53].

The static Fe Kα and Kβ XES spectra of [Fe(bmip)₂](PF₆)₂ and the model complexes were measured at the SSRL beamline 6-2. Samples were measured as powders. Monochromatic incident X-rays at 7.3 keV (double-crystal Si(311) monochromator, 0.2 eV FWHM bandwidth) were used. Identical spectrometers to the LCLS experiment were used. The monochromator energy was calibrated with a Fe foil and the spectrometers calibration was done using elastically scattered X-rays. [Fe(bpy)₃]³⁺and FePc Kβ XES spectra were taken from ref. [10].

**Theory**. Kα XES spectra were calculated at four [Fe(bmip)₂]²⁺ geometries. The geometries for the ground state energy minimum and the ³MC energy minimum were obtained from ref. [46], where optimization was done at PBE0/6-311 G(d,p) level and included MeCN solvent through the polarizable continuum model (Supplementary Note 19). Two other geometries were created on each side of the ³MC energy minimum by changing the geometry along the GS-³MC coordinate, with ΔR = −0.016 Å and ΔR = +0.026 Å. All structures belong to the D₂ point group but as the iron is six-coordinated, Oₕ point group labels will be used to describe the metal orbitals. The spectra were calculated using the *ab initio* Restricted Active Space (RAS) approach in the OpenMolcas package (version v18.0.o180105-1800)[69], using a new efficient configuration interaction algorithm[70]. The electronic structure of the valence and core-ionized electronic states was evaluated at the RASSCF/ANO-RCC-VDZP level of theory[71–73]. The valence active space consisted in 10 electrons distributed in 10 orbitals, the five Fe 3d dominated orbitals and five corresponding correlating orbitals. First, two filled Fe-ligand σ-bonding orbitals together with two metal-dominated $e_g$ orbitals forming σ* orbitals with the ligand. Second, three filled metal-dominated non-bonding $t_{2g}$ orbitals together with three empty Fe 4d orbitals of $t_{2g}$ symmetry. These orbitals were placed in the RAS2 space, where all possible excitations were allowed. The Fe 1s orbital was placed in the RAS3 space, allowing for a maximum of one electron while the three Fe 2p orbitals were placed in the RAS1 space, allowing for a maximum of one hole. To model the core-ionized states and ensure the hole stayed in the core orbitals instead of higher-lying orbitals, the core hole orbitals were kept frozen in the RASSCF optimizations of the 1s and 2p core-ionized states. The valence electronic states were calculated with the singlet and triplet spin multiplicities, while the core-ionized states were calculated with doublet and quartet spin multiplicities. The RASSCF wavefunction optimizations were performed using the state average formalism with equal weighting. For the ³MC states, the 1s core hole was averaged over nine doublet and six quartet states. For the 2p core-hole states 50 doublet and 18 quartet states were used. Spin-orbit coupled states were obtained using a Douglas–Kroll-Hess (DKH) Hamiltonian and atomic mean field integrals[74,75]. Transition dipole moments between the core-ionized states were calculated with the RAS state interaction (RASSI) method[76]. For comparison to experiment, RAS spectra were broadened with a Gaussian of 0.6 eV (FWHM) and Lorentzian function of 1.55 eV (FWHM). The spectrum at the GS geometry was shifted to align with the experimental emission maximum. The same shift was then used for all calculated spectra. The sensitivity of the results with respect to adding dynamic correlation through second-order perturbation theory was tested for the GS and ³MC geometries[77]. The energy difference between the two emission maxima increased by 0.12 eV, while the emission intensity of the ³MC state increased by 5% relative to that of the ground state. These two effects largely cancel when calculating the intensity at the GS emission maximum.

## Data availability

The time-resolved XSS and XES data underlying Fig. 1c–e are provided as a Source Data file. All relevant data that support the findings of this study are available from the authors on reasonable request.

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

## Acknowledgements

K.K., M.E.R., and K.J.G. acknowledge support from the U.S. Department of Energy, Office of Science, Basic Energy Sciences, Chemical Sciences, Geosciences, and Biosciences Division. Use of the Linac Coherent Light Source (LCLS) and the Stanford Synchrotron Radiation Lightsource (SSRL), SLAC National Accelerator Laboratory, is supported by the U.S. Department of Energy, Office of Science, Office of Basic Energy Sciences under Contract No. DE-AC02-76SF00515. M.L. acknowledges financial support from the Knut and Alice Wallenberg Foundation (grant no. KAW-2013.0020) and Olle Engkvists stiftelse. T.C.B.H., E.B., M.G.L., M.P., K.B.M. and M.M.N. gratefully acknowledge support by the Danish Council for Independent Research under grant no. DFF-4002-00272B. M.I.P., M.M.N., and K.B.M. acknowledge support by the Danish Council for Independent Research under grant no. DFF-8021-00347B. This work has been supported by the Danish Council of Independent Research Grant No. 4002-00272 and the Independent Research Fund Denmark Grant No. 8021-00347B. K.H., K.S.K., M.M.N., P.V., T.B.V.D., M.G.L., M.C. and F.B.H. are highly grateful to DANSCATT for supporting the beamtime activities. S.E.C. gratefully acknowledges funding from the Helmholtz Recognition Award. The ELI-ALPS project (GINOP-2.3.6−15-2015-00001) is supported by the European Union and co-financed by the European Regional Development Fund. J.U. acknowledges support from the Knut and Alice Wallenberg Foundation and from the Carl Tryggers Foundation. S.K. acknowledges the support from Knut & Alice Wallenberg foundation (KAW 2014.0370). D.S.S., E.B. and G.V. acknowledge support from the 'Lendület' (Momentum) Program of the Hungarian Academy of Sciences (LP2013-59), the Government of Hungary and the European Regional Development Fund under grant No. VEKOP-2.3.2-16-2017-00015, the European Research Council via contract ERC-StG-259709 (X-cited!), the Hungarian Scientifc Research Fund (OTKA) under contract K109257 and the National Research, Development and Innovation Fund (NKFIH FK 124460). Z.N. acknowledges support from the Bolyai Fellowship of the Hungarian Academy of Sciences.

## Author contributions

K.K. and K.J.G. wrote the manuscript with input from all the authors. K.S.K., T.C.B.H., P.C., K.H., K.B.M., M.M.N., G.V., K.W., V.S., P.P. and J.U. conceived of the experiment. T.C.B.H., K.S.K., K.H., T.B.V.D., K.K., E. Biasin, P.C., R.W.H., M.E.R., M.G.L., F.B.H., P.V., M.C., Z.N., D.S.S., E. Bajnóczi, M.P., R.A.M., J.M.G., S.N., M.S., D.S., H.T.L., S.C., M.M.N., G.V. and J.U. carried out the LCLS experiment. K.K., S.K. and M.E.R. carried out the SSRL experiments. K.K. and K.S.K. analyzed the XES data. K.H., K.K., L.S. and M.G.L. analyzed the XSS data. J.U. measured and K.K. analyzed the optical transient absorption data. M.V., M.L., E.K. and M.D. carried out the RASSCF calculations. E. Biasin carried out the molecular dynamic simulation. Y.L., H.T., C.T. and K.W. synthesized the sample.

## Competing interests

The authors declare no competing interests.
