## [Peer Review File · Nature Communications]

Reviewers' comments:

Reviewer #1 (Remarks to the Author):

The authors investigated the photoinduced electron transfer dynamics of $[\text{Fe}(\text{bmip})_2]^{2+}$ in acetonitrile solution using time-resolved X-ray emission spectroscopy (TR-XES) and time-resolved wide angle X-ray scattering (TR-WAXS). The results from two complementary tools unveiled 1) the 3MC wave packet motion with the time constant of 278 fs and 2) the MLCT to 3MC electron transfer with the time constant of 110 fs.

The overall data reduction procedure and the level of simulation seem suitable for their final arguments. I can hardly find any serious logical jump throughout the reasoning of both TR-WAXS and TR-XES. Besides, these two tools show clear accordance in conclusion. Finally, the target itself is worth being studied considering its potential as a photosensitizer. Therefore, I strongly recommend the publication of this manuscript if the authors can clarify some ambiguities regarding the detailed analysis.

1. The term "wide-angle X-ray scattering (WAXS)" can confuse readers. WAXS was coined to distinguish itself from SAXS (small-angle X-ray scattering) for macromolecules. In other words, WAXS has a relevant meaning with respect to macromolecules. Therefore, instead of TR-WAXS, I suggest time-resolved X-ray solution scattering (TRXSS) or time-resolved X-ray liquidography (TRXL).

2. In page 10 line 320, the authors have assigned the 500 fs decay to the vibrational cooling, which is faster than usual values (~ 10 ps) from similar studies. Also, it does not coincide with the values that can be approximated from the classical diffusion theory of Landau and Lifshitz. Further qualitative reasoning for the conclusion should be provided.

3. In Fig. 5C, the simulations and the experimental data look quite different from each other. While slight differences in low- q regions (up to $1.5\sim 2.0 \text{ \AA}^{-1}$) are relatively understandable, the peak positions in high- q range that encodes the structural information must be taken into account with certain accuracy. I suggest, for example, releasing more degree of freedom in molecular structure to fully make use of the strengths of the X-ray solution scattering data.

4. Among the biexponential dynamics toward the ground state, the faster one with the time constant of 1.5 ps was assigned to $(\text{MLCT}^*) \rightarrow 3\text{MC} \rightarrow 1\text{GS}$. However, according to the TA paper the authors cited [38], this kinetics was not visible not only in the excited state absorption but also in the ground state bleaching, in contrary to the fact that the ground state population must be changed even if it is assumed that the authors' assignment is correct. I suggest giving a plausible explanation for such disagreement between the two studies.

5. In the supplementary information, Fig. S28 is missing.

6. The authors argue that in the SVD, only the highest 3 components are above the noise level. Then, I wonder whether it is mathematically reasonable to extract 6 time-dependent components from the global analysis using the same data set. Additionally, I am satisfied by the fit quality shown in Fig. S29 considering the clear discrepancies in Fig. 5(C).

7. The reported relaxation kinetics of the optical Kerr effect (OKE) in acetonitrile last far longer than the IRFs, and therefore should not be fit to a simple Gaussian function.

8. In section 15 of the SI, the authors mentioned that the solvent heat signal can be described by two rise constants. I wonder if the authors just phenomenologically exclude those for the analysis by fitting the time axis and the authors have ever thought of a physical origin for each of them. I suggest that the authors can add a relevant discussion.

9. In page 4 line 108, I recommend to cite previous studies (on $[\text{Fe}(\text{bimp})_2]^{2+}$ complex) in “contrary to prior ultrafast XES measurements, the $K\alpha$ XES signal shows the signature of vibrational wavepacket dynamics”.

10. It is required to state standard deviations for the structural parameters fitted in the overall TR-WAXS analysis. Also, a typo: Tabel 1 → Table 1.

11. In page 10 line 340, the subtitle “Origin of the XES oscillations” seems to exaggerate the contents of the paragraph as no comments were given regarding the oscillatory motion that might exist in the 3MLCT state.

12. The detailed explanations for each subfigure lack in the caption of Fig. S31.

13. It is better to use equal offsets for each scattering curve. For example, the top two curves of Fig. S30 may distract readers even though the MLCT* and the OKE signal do not have any real physical relationship.

14. The citation for previous time-resolved X-ray solution scattering is too much biased toward their own work (more than 10 self citation versus 1 citation of Ihee group as an example). This should be improved by citing the works from other groups in proportion to the quantities of published works.

Reviewer #2 (Remarks to the Author):

Report on manuscript “Origin of vibrational wavepacket dynamics...” by Kuunus et al.

This manuscript contains a fs WAXS/XES experiment on solvated $\text{Fe}(\text{bimp})_2$ photosensitizer ions, where both tools show a 278 fs period oscillation, identified as the Fe-ligand stretching mode on a

3MC excited state. This is rationalized by a branching of the initially photoexcited reactant in the MLCT manifold to the (vibrationally active) MC states, while a larger fraction undergoes internal conversion into the lowest excited 3MLCT manifold, which is not vibrationally active.

Experimentally, this is certainly groundbreaking work, employing both WAXS and XES in a femtosecond experiment, and utilizing each tool for the analysis and interpretation, which nicely merges into one picture. The analysis utilizes all aspects of data analysis and theoretical modeling, including structural models for WAXS, and RASSCF calculations for the XES, and the observed wavepacket dynamics are just beautiful. As such it warrants publication in this journal.

For a better understanding by the reader I would suggest the authors to go over their assessments once more, and I outline a few key points I noticed: the reactant Fe(bmip)₂ has an optical absorption strength of about 1000 / (M cm) in Fig. 1 of the main text. Therefore I would estimate for the OD shown in the SI

$OD(50 \text{ } \mu\text{m}, 20.3 \text{ mM}, 400\text{nm}) = 0.005\text{cm} \times 0.0203\text{mM} \times 1000/(\text{M cm}) = 0.1015$, and not 0.59

This will become relevant when estimating the photoexcitation yield, which – in the analysis (and further presented in the SI) – was determined to lie in the 74-94 % range. I would suggest to clarify these issues (or possibly correct), also my idea that 45 mJ/cm² corresponds to 4 uJ pulse energy. In order to achieve such a high excitation yield I would expect a laser fluence, which corresponds to several photons pointing to a single absorption cross section, and clarification of these fluence issues would make the presentation more clear

The XES analysis is entirely governed by the suitability of the correction applied to the falsely aligned 4th XES crystal yielding a distorted GS spectrum and subsequent transient spectra, as demonstrated in the SI. This is an important correction, and should therefore be mentioned in the main text (while the treatment itself can remain in the SI). I have to admit that I cannot really imagine how overlapping energy regions in the pumped spectra can be so reliably “resorted” in the analysis, but this may be explainable.

Apart from the need to clarify the experimental conditions, I applaud the authors to this truly pioneering study of the excited state pathways in unusual transition metal complexes.

Reviewer #3 (Remarks to the Author):

This work presents an experimental investigation of the ultrafast dynamics following the photoexcitation of a Fe³⁺ complex. The dynamics of the initially produced MLCT(1) state is followed at three different levels simultaneously in the experimental setup, namely K_α and K_β XES and WAXS.

On the basis of the combined analysis of the three signals and computational modeling of the x-ray spectra, the authors put forward a kinetic model that explains the short-time dynamics and subsequent relaxation, as well as the origin of the sensitivity of the XES signal to the wavepacket dynamics in the valence states.

The latter is very illustrative. Following the conclusions of this work, it becomes a simple matter to pre-compute, for a given system, how sensitive a TR-XES spectrum will be to structural wavepacket dynamics.

The paper is clearly and concisely written. The steps of the analysis leading to the conclusions are explained in much detail in the main paper and SI. Although I am not an experimentalist, I have the impression that this level of detail renders the various steps reproducible. Also, the theoretical results are explained with a sufficient level of detail and are performed at a high level of theory.

The paper will likely promote the further application of XES as a sensitive tool to follow structural dynamics in the fs time-scale and potentially influence future work.

I have a few questions mostly related to mechanistic aspects of the considered process. The paper should, in my opinion, be published, provided these can be convincingly addressed.

Questions:

(1) Maybe I missed the point but I wonder if the authors considered the direct decay MLCT(3) → GS pathway as a possibility in their kinetic models. In Fig. 7 the MLCT(3) → MC(3) channel is marked with a 9ps time-constant, but it could equally, at least in part, go down to GS directly with a good FC overlap.

(2) The MC(3) state decays with a time-constant of 1.5ps to the GS. According to the scheme of energy levels in Fig. 7, though, it would have a sharp crossing with the GS at a quite elongated geometry, such that the intersystem crossing through this channel might display some of the 278fs oscillatory dynamics as well. Is this something that could be maybe buried in the WAXS signal or somewhere else?

(3) The paper alludes to Jahn-Teller effects in the MC(3) state after photoexcitation, which is the consequence of the high symmetry of the complex. Yet the scheme in Fig. 7 refers only to one single (effective?) Fe-L bond length.

How is this considered in the analysis? For example, are the XES spectra calculated for geometries with equal elongation of all Fe-L bonds, or along a symmetry coordinate? The computational details mention that one of the geometries correspond to the MC(3) minimum. Are the Fe-L distances all equal or different in this geometry?

Does this make a difference for the vibronic coupling responsible for the XES sensitivity?

(4) Following up on a similar token, I presume there must be energy values along the corresponding valence and core-excited PES for the various states involved. These could be used in conjunction with Fig. 7 or in the SI to further support the kinetic model. How do these PES look like?

Reviewer #1

Comment 1. The term “wide-angle X-ray scattering (WAXS)” can confuse readers. WAXS was coined to distinguish itself from SAXS (small-angle X-ray scattering) for macromolecules. In other words, WAXS has a relevant meaning with respect to macromolecules. Therefore, instead of TR-WAXS, I suggest time-resolved X-ray solution scattering (TRXSS) or time-resolved X-ray liquidography (TRXL).

Response. We thank the reviewer for the suggestion, but after careful deliberation, we have decided to stay with the TR-WAXS term, as it accurately describes the experimental methodology applied which is, indeed, scattering acquired in a wide angular range (as opposed to SAXS). As such, the term applies equally no matter whether the sample under study is a protein solution or a solution of smaller molecules, as in the present case.

Comment 2. In page 10 line 320, the authors have assigned the 500 fs decay to the vibrational cooling, which is faster than usual values (~10 ps) from similar studies. Also, it does not coincide with the values that can be approximated from the classical diffusion theory of Landau and Lifshitz. Further qualitative reasoning for the conclusion should be provided.

Response. The damping of the oscillations in the signal likely results from both elastic and inelastic scattering, but the measurement is not able to differentiate intramolecular vibrational redistribution from elastic dephasing. We do agree with the reviewer that vibrational cooling to the solvent is not the likely source of a 500 fs time constant process. These details were overlooked in the submitted manuscript because we found that neither the WAXS nor the XES measurements are sensitive to the particular damping mechanism (section 12 in the SI). Analysis of the UV/Vis transient absorption data does have a 20 ps time constant component that likely does result from vibrational cooling of the hot ground state to the solvent (section 6 in the SI).

We have modified the text to more accurately describe what is, and is not, known based on the measurement (line 321): “We assign the damping of the coherent wavepacket motion, characterized by a 500 fs time constant, primarily to dephasing caused by quasi-elastic scattering. Likely, intramolecular vibrational energy distribution also contributes to the damping, but vibrational cooling to the solvent should not be a dominant cause of damping because vibrational cooling occurs on the few-to-many picosecond time scale in MeCN.” Additionally, we replaced the Fig. 5B with a figure that corresponds to an elastic dephasing process.

Comment 3. In Fig. 5C, the simulations and the experimental data look quite different from each other. While slight differences in low-q regions (up to 1.5~2.0 Å⁻¹) are relatively understandable, the peak positions in high-q range that encodes the structural information must be taken into account with certain accuracy. I suggest, for example, releasing more degree of freedom in molecular structure to fully make use of the strengths of the X-ray solution scattering data.

Response. We agree with the reviewer that the match between the data and the extracted solute signal in the $Q = 2.5 \text{ \AA}^{-1}$ lacks the accuracy that can be achieved with WAXS measurements. The dominant reason for this discrepancy results from the lack of robust model for the changes in the solute-solvent pair distribution function (‘solvent cage’) signal. Further adjustment of the solute structure would likely reduce the difference between the data and the model, but is not warranted without a robust solvent cage model. We do think the similarity between the experimental difference signals and the difference signals (Fig. 5C) predicted from

calculated DFT structures for the ^3MC state from Fredin et al. (J. Phys. Chem. Lett. 5, 2014, 2066–2071) provides confidence that the qualitative structural analysis of the WAXS measurement is robust. Specifically, we believe that the scattering data shows reliably a significant expansion of the solute Fe-ligand bond length and its oscillation (consistent with the ^3MC state) but cannot be used to quantify the Fe-ligand bond length in the ^3MC excited state.

We have added the following text to make this clear in the manuscript: “At present, we do not have a robust model for changes in the solute-solvent pair distribution function. This precludes the quantitative extraction of the Fe-ligand bond length from the WAXS signal, but does not preclude the qualitative conclusion that significant Fe-ligand bond expansion and oscillation does occur consistent with motion on the ^3MC excited state potential energy surface.”

We have also added the following sentence to the description of Fig. 4 for clarification: “Since the difference scattering components used in the global analysis come from the SVD of the WAXS data, this does not represent a fit to a specific structural model for the solute or the solute-solvent pair distribution function.”

Comment 4. Among the biexponential dynamics toward the ground state, the faster one with the time constant of 1.5 ps was assigned to (MLCT*)- \rightarrow ^3MC - \rightarrow 1GS. However, according to the TA paper the authors cited [38], this kinetics was not visible not only in the excited state absorption but also in the ground state bleaching, in contrary to the fact that the ground state population must be changed even if it is assumed that the authors’ assignment is correct. I suggest giving a plausible explanation for such disagreement between the two studies.

Response. Most importantly, when the complex is excited at 400 nm and probed in the UV-Visible, we can fit this data to the same kinetic parameters used to model the XES data. This is discussed in Section 6 of the SI. Regarding the specific question of the reviewer, we believe the discrepancy has two primary origins. (1) On the picosecond time scale, it is challenging to differentiate solvation and IVR sources of spectral change from population dynamics in the UV-visible pump-probe measurement; (2) The higher photon energy used on our experiment (3.1 eV) compared to the UV-visible pump-probe measurement (2.55 eV) should influence the branching ratio between MLCT and ^3MC states. This helps explain why the ^3MC channel would be weaker and difficult to isolate in the UV-Visible pump-probe measurement. In contrast, XES and WAXS have specific sensitivity to spin state changes and metal-ligand bond elongation, making the ^3MC the dominant signal source. Note that our analysis of the TA data with 400 nm pump wavelength (Section 6 in the SI) shows rather complex multiexponential dynamics. Without the knowledge from the XES experiment the assignment of these time scales would very challenging.

We have modified the text in the Discussion to clarify these points: “Importantly, when the complex is excited at 400 nm and probed in the UV-Visible, the data can be fit with the same kinetic parameters used to model the XES data (SI). This supports the conclusion that the branching ratio between $^3\text{MLCT}$ and ^3MC formation depends on the pump photon energy.”

Comment 5. In the supplementary information, Fig. S28 is missing.

Response. We thank the reviewer for pointing this error in the PDF conversion, it has now been corrected.

Comment 6. The authors argue that in the SVD, only the highest 3 components are above the noise level. Then, I wonder whether it is mathematically reasonable to extract 6 time-dependent

components from the global analysis using the same data set. Additionally, I am satisfied by the fit quality shown in Fig. S29 considering the clear discrepancies in Fig. 5(C).

Response. Mathematically, there is no upper limit to the number of time-dependent components one can use for a given number of SVD components. For example, there could be N number of time-scales in a data with no evolution in spectral shape (only amplitude change). The time evolution of such data would be described by a single above-the-noise SVD component. Then, one must use N time-dependent components with the same spectral shape (but different amplitude) to describe the dynamics of the data. In the current data, the ^3MC and the oscillatory difference signals have a very similar shape (but very different dynamics). Apparently, the shape of the MLCT* and the fast residual signals (this was previously labelled as the OKE signal, see the next comment) can be reasonably well described with a linear combination of $^3\text{MLCT}$, ^3MC and Heat signals. That is not so surprising, given that both signals are very small and lack distinct characteristics.

Note that the global fitting results in Fig. S30 do not include the simulated ^3MC signal in Fig. 5C (labelled as “sim.”), but rather the experimental ^3MC signal extracted from the global fit (labelled as “exp.”). To avoid this confusion, we changed the text in the caption of Fig. 5: “(C) Simulation of the time-dependent ^3MC difference signal and comparison to the experiment.” to “(C) Simulation of the time-dependent ^3MC difference signal and comparison to the experimental ^3MC signal extracted from the global fit (including the oscillatory component).” In the main text (line 328) we replaced: “Comparison of the resulting simulated ^3MC difference WAXS signals with the experiment shows ...” with “Comparison of the resulting simulated ^3MC difference WAXS signals with the experimental ^3MC signal extracted from the global fit shows ...”

Comment 7. The reported relaxation kinetics of the optical Kerr effect (OKE) in acetonitrile last far longer than the IRFs, and therefore should not be fit to a simple Gaussian function.

Response. We agree with this statement of the Reviewer. We included now an analysis of the time-dependent OKE signal (Section 10 in the SI) to clarify this and to demonstrate agreement with the previously published results. Also, we believe that previous assignment of the ultrafast residual signal in the isotropic scattering to OKE was not correct. We believe that this residual signal is more likely a result of solvent response to the new charge distribution in the MLCT state, an effect we are pursuing in other ongoing studies but where no final results have yet been achieved. Note that the interpretation of this ultrafast residual signal does not influence the conclusions based on the solute dynamics at later time scales (>200 fs) in any way. We changed the label of the corresponding time-dependent component in the SI (Fig. S29 and S31 and formula in the section 16). We replaced in the main text “We find that the <200 fs structural dynamics cannot be captured by the population dynamics of MLCT*, likely due to a small contribution from the MeCN optical Kerr effect signal (SI),” with “We find that the <200 fs structural dynamics cannot be captured by the population dynamics of MLCT*, due both solvent and solvation contributions to the difference signal that we have not included in the structural model (SI).”

Comment 8. In section 15 of the SI, the authors mentioned that the solvent heat signal can be described by two rise constants. I wonder if the authors just phenomenologically exclude those for the analysis by fitting the time axis and the authors have ever thought of a physical origin for each of them. I suggest that the authors can add a relevant discussion.

Response. Yes, time-dependence of the solvent heating was established by fitting of the solvent heat signal at each time delay. We included now a more thorough description of this in

the SI (Section 11), along with a new figure (Fig. S27). We did not subtract the solvent heat signal before the global fitting, but rather included it in the global fit as one of the time-dependent components.

We believe that the two observed heating time constant are phenomenological and not straightforward to interpret. They can be qualitatively assigned to impulsive heating (initial expansion and collision with wall of solvent cage, as well as from OKE relaxation) and a combination of vibrational cooling and non-radiative excited state decay processes. As the Reviewer suggested, we included a short discussion about this in the SI (Section 11): “The time-dependence of the solvent heat amplitude was analyzed by fitting the solvent heat signal at each time delay (Fig. S27B). We found that solvent heating can be well described with two exponential rise times corresponding to 0.35 ps and 13.2 ps. These phenomenological time constants are a convolution of multiple time scales related to solute internal conversion (MLCT* \rightarrow ³MLCT \rightarrow GS and MLCT* \rightarrow ³MC \rightarrow GS) and vibrational energy distribution (IVR) processes, as well as fast solvation (<1 ps) and slower vibrational cooling time scales of the various solute electronic state populations (~10 ps). In addition, direct excitation of solvent via OKE process contributes to ultrafast solvent heating (Fig. S25). The biexponential fit in Fig. S27B is directly used in global fitting of the whole isotropic WAXS data to model the time-dependence of the solvent heat signal (see below).”

Comment 9. In page 4 line 108, I recommend to cite previous studies (on [Fe(bimp)₂]²⁺ complex) in “contrary to prior ultrafast XES measurements, the K α XES signal shows the signature of vibrational wavepacket dynamics”.

Response. This is a misunderstanding due to our poor phrasing of the sentence. With prior studies we had in mind prior studies on transition metal complexes in general (not on the [Fe(bimp)₂]²⁺ complex, since the current investigation is the first XES experiment on this complex). We changed the sentence from: “Contrary to prior ultrafast XES measurements, the K α XES signal shows the signature of vibrational wavepacket dynamics,” to “Contrary to prior ultrafast XES measurements on transition metal complexes [10-14], the K α XES signal of [Fe(bimp)₂]²⁺ shows the signature of vibrational wavepacket dynamics.”

Comment 10. It is required to state standard deviations for the structural parameters fitted in the overall TR-WAXS analysis. Also, a typo: Tabel 1 -> Table 1.

Response. The structural parameters in Table 1 are not from a fit, but from a DFT calculation in Ref. 47 as discussed in detail in our response to comment 3. As such, we cannot associate any standard deviations to these.

Comment 11. In page 10 line 340, the subtitle “Origin of the XES oscillations” seems to exaggerate the contents of the paragraph as no comments were given regarding the oscillatory motion that might exist in the 3MLCT state.

Response. Indeed, the aim of this subsection is not so much to explain the origin of the oscillatory nuclear wavepacket dynamics, but to explain the sensitivity of XES to this dynamic. To make this point clearer we changed the title of the subsection to “Origin of the XES sensitivity to nuclear wavepacket oscillations”.

Comment 12. The detailed explanations for each subfigure lack in the caption of Fig. S31.

Response. We included now the descriptions of subfigures in the caption.

Comment 13. It is better to use equal offsets for each scattering curve. For example, the top two curves of Fig. S30 may distract readers even though the MLCT* and the OKE signal do not have any real physical relationship.

Response. We followed the advice of the Reviewer and plotted each scattering signal at equal offset levels (Fig. S31).

Comment 14. The citation for previous time-resolved X-ray solution scattering is too much biased toward their own work (more than 10 self citation versus 1 citation of Ihee group as an example). This should be improved by citing the works from other groups in proportion to the quantities of published works.

Response. We thank the reviewer for this comment, which we should have been addressed in the initial submission. We have modified our citations to better reflect the full field of research.

Reviewer #2

Comment 1. For a better understanding by the reader I would suggest the authors to go over their assessments once more, and I outline a few key points I noticed: the reactant Fe(bmip)₂ has an optical absorption strength of about 1000 / (M cm) in Fig. 1 of the main text. Therefore I would estimate for the OD shown in the SI $OD(50 \text{ } \mu\text{m}, 20.3 \text{ mM}, 400\text{nm}) = 0.005\text{cm} \times 0.0203\text{mM} \times 1000/(\text{M cm}) = 0.1015$, and not 0.59. This will become relevant when estimating the photoexcitation yield, which – in the analysis (and further presented in the SI) – was determined to lie in the 74-94 % range. I would suggest to clarify these issues (or possibly correct), also my idea that 45 mJ/cm² corresponds to 4 uJ pulse energy. In order to achieve such a high excitation yield I would expect a laser fluence, which corresponds to several photons pointing to a single absorption cross section, and clarification of these fluence issues would make the presentation more clear

Response. We thank the reviewer for their meticulous review. The problem proves to be a mislabeled y-axis in Fig. 1A. The correct units for molar absorptivity in Fig. 1A are 10⁴/cm/M (not 10³/cm/M as it was shown before). Consequently, the cross-section is a factor of 10 larger than the incorrect plot shown in the submitted manuscript. In addition, the spectrum we showed did not correspond to the [Fe(bmip)₂]²⁺ molecule, but to [Fe(btbp)₂]²⁺ instead (latter molecule is discussed in section 8 of the SI where we used it to estimate the excitation yield). We are very thankful for the reviewer for pointing out this error. We corrected the Fig. 1A and our spectrum now agrees with the previously published spectrum in Ref. [43].

Using the correct spectrum the expected OD is $0.005\text{cm} \times 0.0203 \text{ M} \times 5800/(\text{M cm}) \approx 0.59$, in an agreement with the OD measured directly for the solution used during the experiment.

Regarding the high excitation yield, this reflects the challenging need to extract useful information from brief XFEL experiments. Fortunately, our experience with Fe²⁺ coordination complexes has been these molecules tolerate high fluence with minimal non-linear absorption. In the SI (section 8), we showed that a very similar complex [Fe(btbp)₂]²⁺ under identical experimental conditions exhibited excited state dynamics agreeing with the previous experiments (Ref. [3] in the SI). Also, we carried out lower fluence TA experiments on [Fe(bmip)₂]²⁺ (Section 5 and 6 in the SI) and we found that these are consistent with the X-ray results after subtracting the 4% of excited molecules that are two-photon ionized. As we argued in the Section 8 of the SI, we believe that this is because the dominant non-linear effect is saturated absorption.

Comment 2. The XES analysis is entirely governed by the suitability of the correction applied to the falsely aligned 4th XES crystal yielding a distorted GS spectrum and subsequent transient spectra, as demonstrated in the SI. This is an important correction, and should therefore be mentioned in the main text (while the treatment itself can remain in the SI). I have to admit that I cannot really imagine how overlapping energy regions in the pumped spectra can be so reliably “resorted” in the analysis, but this may be explainable.

Response. We expanded now our description of the XES correction procedure and we also provide an explanation of the procedure we have adapted. The text in Section 2 of the SI now reads:

“Misalignment of one of the four analyzer crystals of the multicrystal spectrometer during the LCLS experiment resulted in distorted K β XES spectra. In order to remove this distortion, we

express the measured (uncorrected) spectrum S_{uncorr} as a sum of two correct spectra S_{corr} that are shifted with respect to each other:

$$S_{\text{uncorr}}(x) = (1 - r)S_{\text{corr}}(x) + rS_{\text{corr}}(x - s)$$

Here, r is the relative intensity of the misaligned (shifted) spectrum and s is the shift. Following from the properties of the δ -function, we can express S_{uncorr} as a convolution

$$S_{\text{uncorr}}(x) = S_{\text{corr}}(x) * D(x)$$

where S_{corr} is convolved with a function D describing the misalignment:

$$D(x) = (1 - r)\delta(x) + r\delta(x - s)$$

Misalignment correction is therefore equivalent to a deconvolution problem and can be readily solved by applying the convolution theorem. Correct spectrum can be thus calculated from a measured (uncorrected) spectrum with a following formula

$$S_{\text{corr}}(x) = FT^{-1} \left\{ \frac{FT\{S_{\text{uncorr}}(x)\}}{FT\{D(x)\}} \right\}$$

Direct application of the convolution theorem to carry out the deconvolution is possible here because D is a sum of δ -functions and is noise free. Therefore, singularities do not appear in the division with $FT\{D\}$. In general, deconvolution of noisy data requires regularization (e.g. by imposing some smoothness criterium to the result), this is fortunately not necessary here (due to the specific form of D). The correction procedure includes finding appropriate relative intensity r and shift s of the two delta peaks in D . This is achieved by comparison to the known $[\text{Fe}(\text{b mip})_2]^{2+}$ ground state spectrum measured at the SSRL with an identical spectrometer. In Fig. S3 we demonstrate the application of this procedure to the ground state $\text{K}\beta$ XES spectrum. We found that the misaligned portion of the spectrum is shifted by 20 pixels and it corresponds to 28% of the total intensity. Exactly the same D as shown in Fig. S3 was then applied to correct all the time-resolved XES spectra at different time delays. This is valid because the correction procedure defined above is a linear operation and does not depend on the shape of S_{uncorr} . Therefore, it can be applied to spectra with any shape (i.e. not only the ground state spectrum, but also to all the transient spectra). Also, it is evident from the power spectrum of D , $FT\{D\} * FT\{D\}$ in Fig. S3 that this procedure does not involve any filtering (e.g. low pass) or smoothing of the data. We demonstrate latter by comparing the residuals of S_{corr} taken with respect to the reference spectrum and the residuals of S_{uncorr} taken with respect to a “distorted” reference spectrum (Fig. S4).”

Reviewer #3

Comment 1. Maybe I missed the point but I wonder if the authors considered the direct decay MLCT(3) → GS pathway as a possibility in their kinetic models. In Fig. 7 the MLCT(3) → MC(3) channel is marked with a 9ps time-constant, but it could equally, at least in part, go down to GS directly with a good FC overlap.

Response. Indeed, the reviewer makes a good point. As we also described in the discussion section, experimentally we cannot distinguish between the direct ³MLCT→GS and indirect ³MLCT→³MC→GS decay pathways. This is because the ³MC lifetime (1.5 ps) is too short compared to the ³MLCT lifetime (9 ps). We showed that also in Section 9 in the SI (populations in Fig. S23C and Fig. S24C are very similar and the differences can be accommodated in the fit by adjusting the spectral intensities). However, in Fig. 7 we are favoring the indirect pathway because the ³MC state has been shown to facilitate the ³MLCT decay in similar complexes (Ref. 14 and 51) and this has been supported also by the calculations (Ref. 47 and 48).

To clarify this issue we have modified the Discussion to read, “Because [Fe(bmip)₂]²⁺ has a slightly longer ³MLCT and a slightly shorter ³MC lifetime than [Fe(bpy)(btz)₂]²⁺, we cannot experimentally distinguish between ³MLCT→³MC→GS or ³MLCT→GS relaxation pathways (see SI).”

Comment 2. The MC(3) state decays with a time-constant of 1.5ps to the GS. According to the scheme of energy levels in Fig. 7, though, it would have a sharp crossing with the GS at a quite elongated geometry, such that the intersystem crossing through this channel might display some of the 278fs oscillatory dynamics as well. Is this something that could be maybe buried in the WAXS signal or somewhere else?

Response. The energy difference between the ³MC and the GS states in Fig. 7 is not to scale with respect to the displacement of the ³MC surface (that is why we didn't include an energy scale in the y-axis). Thus, the crossing point of the ³MC and GS should have been viewed solely from a qualitative perspective. To eliminate this confusion, we modified Fig. 7 by removing the upper right part of the GS surface. In the caption of Fig. 7 we also mention that the surfaces are qualitative to avoid confusing the reader.

We think it should not be possible to observe wavepacket dynamics on the GS surface because the GS population time (1.5 ps) is significantly longer than the oscillation period (278 fs) and that should “wash out” any oscillatory dynamics.

Comment 3. The paper alludes to Jahn-Teller effects in the MC(3) state after photoexcitation, which is the consequence of the high symmetry of the complex. Yet the scheme in Fig. 7 refers only to one single (effective?) Fe-L bond length. How is this considered in the analysis? For example, are the XES spectra calculated for geometries with equal elongation of all Fe-L bonds, or along a symmetry coordinate? The computational details mention that one of the geometries correspond to the MC(3) minimum. Are the Fe-L distances all equal or different in this geometry? Does this make a difference for the vibronic coupling responsible for the XES sensitivity?

Response. Yes, the coordinate in Fig. 7 is an averaged Fe-ligand bond length which is most relevant for a qualitative representation of the origin of the oscillatory signal in the XES. As we argue in the paper, the dominant character of this coordinate is Fe-ligand stretching. However, in the calculations of the XES and scattering signals the structural changes correspond to those predicted for the transition from GS to the ³MC state potential energy surface. Specifically, the

calculations were done using the structures that are on the coordinate connecting the equilibrium GS and ^3MC geometries (Table 1, full list of coordinates are printed in Section 19 in the SI). This was done by defining a vector that connects the GS and the ^3MC structures and subsequently scaling this vector to construct structure that lie between the GS and the ^3MC structures or beyond the ^3MC structure. This takes into account pseudo-Jahn-Teller distortion in the ^3MC (thus, Fe-ligand distances are in general not equal). To describe these constructed structures, we defined a single scalar metric that is the average Fe-ligand bond length and we used that metric throughout the paper. We found that the XES and scattering signals calculated from such structures agree very well with the observed experimental signals. It would be definitely desirable to “map out” the multi-dimensional nuclear space more thoroughly, but because the XES calculations are time-consuming, this has not been achieved yet. This investigation, however, clearly indicates that Fe-stretching mode (that maps well to average Fe-ligand bond length observed in the X-ray scattering) is a relevant coordinate for the XES sensitivity.

Comment 4. Following up on a similar token, I presume there must be energy values along the corresponding valence and core-excited PES for the various states involved. These could be used in conjunction with Fig. 7 or in the SI to further support the kinetic model. How do these PES look like?

Response. Indeed, we have energy values for the 1s and 2p core-excited PESs from the RASSCF calculations used to simulate the XES spectra. However, we haven't calculated valence PESs in this work. These have been calculated in Ref. 47 and 48 and these qualitatively agree with the surfaces presented in Fig 7. As the Reviewer suggested, we included now a figure with the core-excited PESs in the SI (section 15).

REVIEWERS' COMMENTS:

Reviewer #1 (Remarks to the Author):

The authors addressed the concerns raised for the original version of the manuscript. except one point (see below). With the notable improvements regarding the plausibility of kinetic model assignment as well as the overall readability of the paper, the authors clearly stated the limitations of the demonstrated structural analysis.

Nevertheless, the issue with the terminology was not properly resolved and should be rectified prior to the publication. For macromolecules, the terminology "WAXS" makes sense to complement the term "SAXS" in that the latter refers to small scattering angle (Q less than 0.2 \AA^{-1}) whereas the former refers to larger scattering angle (Q larger than 0.2 \AA^{-1}). However, in the case of small molecules, a clear criteria that distinguishes between small and wide angle scattering has not been established. If we consider the maximum Q of typical solution scattering experiments for small molecules which has been reported so far (for example 9 \AA^{-1}), the maximum Q of 3.5 \AA^{-1} for this work is not large enough to be emphasized as "WIDE" angle scattering at all. Hence, the authors should not use the term "WAXS" so that this term can be used in the future when much larger Q values become accessible with the aid of technical improvements. Instead, more generally accepted terms such as "X-ray solution scattering" and "diffuse X-ray scattering" should be used.

It seems there is a typo in Fig. S32: "oscillatory $K\alpha$ XES (B, E, F) ..." should be "oscillatory $K\alpha$ XES (B, E, H) ...".

Hyotcherl lhee

Reviewer #3 (Remarks to the Author):

Dear Editor,

the authors have addressed all raised points convincingly and have made, when needed, the corresponding modifications to the manuscript.

As already mentioned in my initial report, I think that this paper is highly relevant for the field of time-resolved x-ray science and I recommend its publication in NComms.

Reviewer #1

Comment 1. The authors addressed the concerns raised for the original version of the manuscript. except one point (see below). With the notable improvements regarding the plausibility of kinetic model assignment as well as the overall readability of the paper, the authors clearly stated the limitations of the demonstrated structural analysis.

Nevertheless, the issue with the terminology was not properly resolved and should be rectified prior to the publication. For macromolecules, the terminology “WAXS” makes sense to complement the term “SAXS” in that the latter refers to small scattering angle (Q less than 0.2 \AA^{-1}) whereas the former refers to larger scattering angle (Q larger than 0.2 \AA^{-1}). However, in the case of small molecules, a clear criteria that distinguishes between small and wide angle scattering has not been established. If we consider the maximum Q of typical solution scattering experiments for small molecules which has been reported so far (for example 9 \AA^{-1}), the maximum Q of 3.5 \AA^{-1} for this work is not large enough to be emphasized as “WIDE” angle scattering at all. Hence, the authors should not use the term “WAXS” so that this term can be used in the future when much larger Q values become accessible with the aid of technical improvements. Instead, more generally accepted terms such as “X-ray solution scattering” and “diffuse X-ray scattering” should be used.

Response. We thank the reviewer for these comments. With the common interest of avoiding confusion, we have decided to use the term X-ray solution scattering (XSS) throughout the paper. We note that the term diffuse X-ray scattering, which we have used in our previous publications, appears confusing for the solid state community where this term indicates the intensity between the Bragg peaks.

Comment 2. It seems there is a typo in Fig. S32: “oscillatory $K\alpha$ XES (B, E, F) ...” should be “oscillatory $K\alpha$ XES (B, E, H) ...”.

Response. We thank the reviewer for pointing out this typo. We fixed it now.

Reviewer #3

the authors have addressed all raised points convincingly and have made, when needed, the corresponding modifications to the manuscript.

As already mentioned in my initial report, I think that this paper is highly relevant for the field of time-resolved x-ray science and I recommend its publication in NComms.